# Religion and Positive Youth Development: Challenges for Children and Youth with Autism Spectrum Disorder

**Susan Crawford Sullivan** [1,*] and **Victoria Aramini** [2]

1   Department of Sociology and Anthropology, College of the Holy Cross, Worcester, MA 01610, USA
2   Carroll School of Management, Boston College, Boston, MA 02467, USA; aramini@bc.edu
*   Correspondence: ssulliva@holycross.edu

**Abstract:** While previous research confirms the role religion can play in positive youth development, much existing research leaves out consideration of underrepresented populations. One important underrepresented population is children and youth with autism spectrum disorder (ASD), which now impacts one in 59 children in the United States. Using qualitative data collected from in depth, semi-structured, face-to-face interviews of 53 parents/caregivers, in this article, we analyze barriers and opportunities in religious education for children and youth with autism spectrum disorder. We analyze factors impacting whether parents perceive their children to have a supportive and appropriate religious education experience or an unsupportive and alienating experience. We also provide recommendations for congregations and argue for inclusion of children with autism spectrum disorder in future research on religion and positive youth development.

**Keywords:** youth development; autism spectrum disorder; family; congregations; religious socialization

---

## 1. Introduction

Substantial amounts of prior research point to religion's role in positive youth development (Abo-Zena and Barry 2013; Antrop-González et al. 2007; King and Furrow 2004; Smith and Denton 2005). However, much existing research on religion and youth development leaves out youth underrepresented due to race, ethnicity, minority religion, low socioeconomic status, disability, etc. Children and youth with autism spectrum disorder (ASD) make up one such underrepresented and understudied group. In the United States, autism spectrum disorder impacts one in 59 children (CDC 2019). Given that the limited existing research suggests that religion has a positive impact on the development of and quality of life for youth with ASD (Biggs and Carter 2016; Liu et al. 2014), in this article, we take a lifespan perspective and ecological approach to youth development by examining one component of religious socialization—religious education—for such children and youth.

There are several key contributions of this study. We aim to fill gaps in the scholarly literature through in-depth interview research focused on religious education experiences of children with autism spectrum disorder from the point of view of parents/caregivers. Listening to parents provide their perspective of mixed experiences with religious education, we analyze what contributes to religious education being perceived as supportive of children with ASD versus being unsupportive or rejecting. Another contribution of this study is recommendations for congregations, which are drawn directly from parents' voices. Finally, this study expands the literature on religion and positive youth development by using a lifespan model to include children and youth with autism spectrum disorder.

Considering religion and youth development from a lifespan perspective points to religious socialization, that is, how parents and others transmit religion and how youth construct and internalize

it. In order for religion to impact youth development, religious socialization needs to take place. While a comprehensive discussion of religious socialization is beyond the scope of this article, religious socialization includes the processes by which parents and other adults impart or transmit to children a religious tradition and identity, as well as the ways in which youth construct their own religious choices (Bengtson 2013; Pusztai and Demeter-Karászi 2019). Thus, parents and others transmit faith beliefs, practices, and communities to children and youth, from which youth interpret and construct beliefs and practices (Bengtson 2013; Pusztai and Demeter-Karászi 2019). Parents are the most important factor in religious socialization, and children who are raised by parents who are religious are more likely themselves to practice religious faith than children raised in non-religious families (Bengtson 2013; Smith and Denton 2005; Smith et al. 2014). Outside of the family, other factors that contribute to religious socialization include faith communities, religious education, and religious peer groups (Vaidyanathan 2011). Parents rely on the formal mechanism of religious education programming provided by the church, synagogue, or mosque to aid in the formation of children's religious identity and knowledge. Religious education can provide children and youth with information about beliefs and practices, as well as provide role models in religious educators. Parents of children with autism spectrum disorder often find religious socialization of their children outside of the family particularly challenging (Ault et al. 2013a, 2013b; Howell and Pierson 2010; Speraw 2006; Whitehead 2018).

## 2. Autism Spectrum Disorder, Religion, and Youth Development

Autism spectrum disorder, a now-common developmental disorder that begins in early childhood, impairs children's communication and/or social skills. Affected children may engage in repetitive motions, have impaired language development or be non-verbal, be extremely sensitive to light or noise, and/or have difficulty with social cues and interactions. Due to substantial variation in severity and ways in which symptoms manifest, it is called a spectrum disorder (National Institute of Mental Health n.d.). ASD rates have greatly increased in recent decades, and currently one in 59 United States (US) children are estimated to have an autism spectrum disorder, with boys being diagnosed at four times the rate of girls (CDC 2019). In 2012, several autism-related disorders were reclassified under the diagnosis of "autism spectrum disorder"; these include autism, Asperger's syndrome, childhood disintegrative disorder, and pervasive developmental disorder (not otherwise specified) (American Psychiatric Association 2013). Autism spectrum disorder diagnoses in the United States vary by gender, geography, race, ethnicity, and socioeconomic status (Baio et al. 2018). Compared with children and adolescents with attention deficit disorder/attention deficit hyperactivity disorder (ADHD), as well as typically developing youth, children and adolescents with autism spectrum disorder are more likely to not participate in organized activities or attend religious services (Lee et al. 2008). Similarly, compared with both typical controls and children with ADHD, parents report higher levels of quality of life concerns for their children with autism spectrum disorder, such as being bullied and having difficulty coping with stress (Lee et al. 2008).

### 2.1. Religion and Positive Youth Development

Positive youth development can be characterized by the constructs of the "5 C's"—competence, confidence, connection, character, and caring/compassion—leading youth to contribute positively to their communities (Eccles and Gootman 2002; Roth and Brooks-Gunn 2003; Lerner 2005; Lerner et al. 2000). Numerous studies in recent years show the association between religion and positive youth development (for one review, see King and Furrow 2004). Smith and Denton (2005) analyze religion's contribution to positive youth development, including spiritual experiences, which may help solidify moral commitments and positive behaviors, involvement in a supportive community, coping skills, leadership opportunities, increased social and cultural capital, and access to adult role models (241–49). King and Furrow (2004) work highlights the key role of religious-based social capital in mediating religion's beneficial effect on youth development. Studies analyzing positive associations with religion for urban youth of color, including high academic achievement, link religious beliefs and participation

with positive mentoring, positive peer and adult social capital, and personal religious beliefs that foster resilience (Antrop-González et al. 2007; Cook 2000). Religion can contribute positively to development in immigrant youth through social support, volunteering, and source of meaning, as well as through limiting risky adolescent behavior (Abo-Zena and Barry 2013). While the focus in the positive youth development literature is religion's role on developing positive traits, religion is also associated with protectiveness from risky adolescent behaviors. Youth involved in religion report less depression and less alcohol and marijuana use; they are also less likely to engage in risky sexual activity or skip school (Smith and Denton 2005). Immigrant youth also report reduced risky behavior as a benefit to religion (Abo-Zena and Barry 2013).

It is important to note that while there is a substantial body of literature on the association between religion and positive youth development, religion does not always contribute positively. Youth and emerging adults can face spiritual struggles, experience religious discrimination as religious minorities, or draw on religion to discriminate against others (Abo-Zena and Barry 2013; Magyar-Russel et al. 2014). If one perceives one's religion as not accepting of sexual minorities, religion is associated with more negative self-identity and lack of self-acceptance for sexual minority youth (Page et al. 2013).

## 2.2. Religion and Positive Youth Development: Autism Spectrum Disorder

Thus, there is a large body of research on religion and youth development, with much finding positive impact. Notably, while personal religious beliefs provide resources for youth, much research indicates that it is the increased social support and social capital in religious communities that is particularly beneficial for youth development. In general, research on religion and youth development does not address youth with autism spectrum disorder, leaving out a substantial and often-marginalized population of widely varying abilities. The American Association on Intellectual and Developmental Disabilities recognizes religion and spirituality as important parts of the human experience to which people with disabilities should have access as they choose (AAIDD/The Arc 2010). Carter (2007) found similar rates of viewing faith as an important part of their lives in people with and without disabilities. Spiritual and religious beliefs and practices can be important and contribute positively to the quality of life for people with disabilities (National Organization on Disability 2000).

While there is only limited research on the impact of religion or spirituality on youth development for adolescents or young adults with autism spectrum disorder, a 2014 qualitative study of young adults with ASD or intellectual disabilities found that the overwhelming majority said they prayed and noted the positive influence of faith and religion in their lives; just over half participated to some degree in congregational activities, and almost half participated in service to others through their congregation (Liu et al. 2014). The review by Vogel et al. (2006) of the impact of religious involvement on people with intellectual and developmental disabilities found that religion can provide a positive impact on sense of self, sense of belonging, and friendship. These studies point to religion as a resource that can contribute to positive youth development for people with autism spectrum disorder.

Peer interaction and social support are among the biggest challenges for youth with ASD (Biggs and Carter 2016; Robertson 2010). Biggs and Carter (2016) quantitative study surveyed parents of youth with ASD or intellectual disability and found that while the lowest-rated items for both groups were peer and social support, religious faith predicted higher quality of life ratings in social support and peer relations. Although the constructs of positive youth development found in scholarship on typically developing youth are not usually applied to youth with autism spectrum disorder, having a higher quality of life in terms of social support and peer relations is an undeniably positive youth development for this population. Furthermore, the severity of impairment with ASD varies widely, with less-impaired youth able to attend mainstream schooling and attain jobs and, presumably, grow in the 5Cs of positive youth development as described for typically developing youth.

### 2.3. Including Children with ASD in Religious Organizations: Challenges for Families

Despite an increase in religiously unaffiliated adults in the US in recent years, 38% of American adults say in surveys that they attend religious services weekly or almost weekly, and 90% of adults say that they believe in a higher power (Fahmy 2018; Newport 2018). While religious parents seek to share their religious traditions with their children, previous studies show that even parents who are not very involved in organized religion, particularly mothers, report involving their families in religious organizations to benefit their children (Ammerman 1997; Edgell 2006). In churches, mosques, synagogues, and other religious institutions, parents seek community and reinforcement of values. Ammerman (1997) "Golden Rule Christians" wanted to engage with churches in raising their children, even though they may not regularly attend or hold traditional beliefs. In another study, mothers wanted to involve children in churches even if they did not themselves have strong salience of religion in their own lives, to pass on traditions and help children decide more knowledgeably about religion (Edgell 2006). The main agents passing along religious traditions are parents, congregations, religious education, and peer groups (Vaidyanathan 2011). For children with autism spectrum disorder, parents often face challenges in accessing channels of religious socialization outside the family, such as participation in religious congregations and religious education.

A primary barrier is family participation in religious congregations overall, as previous research indicates that families of children with autism spectrum disorder participate in religious organizations (as well as non-religious organizations) at lower rates than families of children without disabilities or families of children with ADHD (Lee et al. 2008). Stressors, such as communication difficulties, lack of needed education, and parents' difficulty in controlling children's behavior (Tarakeshwar and Pargament 2001), as well as stigma and disapproval (Blum 2015; Francis 2015) from others, can make it difficult for parents of children with ASD to participate in community organizations, including religious ones. Whitehead (2018) recent longitudinal study found that children with chronic health conditions that limit communication and social interaction, such as autism spectrum disorder, are most likely to never attend religious services when compared with children with other types of chronic health conditions and children with no condition.

When families of children with autism spectrum disorder do participate in religious congregations, many find it unsatisfactory due to lack of inclusion and welcome for their child (Ault et al. 2013a) and negative reactions to their child's behavior from congregation members (White 2009). Coulthard and Fitzgerald (1999) study of parents of children with ASD found that while most engaged in personal prayer, very few would turn to religious community members for support. Participation in organized religious activities was found to increase negative outcomes and distress for mothers of children with ASD, including higher levels of depression and parenting stress and lower levels self-esteem, well-being, and sense of self-efficacy (Ekas et al. 2009).

There have been relatively few academic studies of parent experiences with religious education for children with autism spectrum disorder. Speraw (2006) small qualitative study of parents/caregivers of children with special needs (some of whom had ASD) found disappointing experiences with religious education, leading to crises of faith. Other studies that include some parents of children with autism spectrum disorder highlight congregational lack of support for children's participation in religious education (Ault et al. 2013a; Baggerman et al. 2015; Jacober 2010). Conversely, a few case studies highlight more positive and satisfactory experiences with religious education for individuals with ASD by analyzing successful adaptive and inclusion models, for example, for Bar Mitzvahs (Goldstein and Ault 2015) or Sunday School (Howell and Pierson 2010).

As reviewed earlier, a noteworthy body of literature finds religion to positively benefit youth development. For youth with autism spectrum disorder, the limited amount of existing research suggests that for some, religion is associated with higher quality of life and positive development (Biggs and Carter 2016; Liu et al. 2014). If religion and spirituality contribute to quality of life and positive youth development for youth with ASD, then it is important to better investigate mechanisms of religious involvement for these youth. In this article, we conduct in-depth interviews with parents and

specifically investigate experiences with religious education, aiming to illuminate the lived experience of families as they seek to involve their children in a congregation's religious learning opportunities. The qualitative data, garnered through in-depth, semi-structured, face-to-face interviews, enable the analysis of factors impacting positive and negative religious education experiences, lead to recommendations for congregations, and point toward a robust research agenda for religion and positive youth development for this often-marginalized diverse population.

## 3. Methods

In this article, we draw from in-depth, semi-structured, face-to-face interviews with 53 parents/caregivers (47 parents and six young adult siblings who helped with caregiving) of children with autism spectrum disorder, as well as face-to-face interviews with two religious educators in a program designed for children with ASD or intellectual disabilities. We also analyze a field visit to a religious education program that was designed for children with ASD or intellectual disabilities.

After obtaining approval for the human subjects study from the first author's college's institutional review board, the second author communicated with ASD support centers in and around a large New England city to recruit parents for a study on religion in the lives of families of children with autism spectrum disorder. The recruitment flyer sent read in part: "Do you have a loved one with an autism spectrum disorder? Are you willing to share your experiences? We want to listen." The flyer indicated that a professor was doing a "project on the faith experiences of families who have members affected by autism spectrum disorder," stating, "No particular religious background is needed–we want to hear from people with a range of experiences with faith and congregations, from none to very involved." An ASD resource professional also posted information about the study on Facebook. Flyers were sent and follow-up phone calls conducted in six ASD support centers, and flyers were also sent to a few local congregations and synagogues. Some of the participants were recruited through snowball sampling building off of contacts, including at a conference for young adult siblings of children with ASD. The initial recruitment began in December 2013, continuing into 2015. The interviews were conducted during a fifteen-month period in 2014 and 2015.

Although most of the respondents were from New England, a project research associate interviewed nine parents in the southern part of the United States, in order to secure more geographic diversity in the study. The focal child(ren) of the parents interviewed ranged from 4 to 29 years old, with a median age of 12. There were more parents/caregivers of boys represented than girls (46 versus seven). Most of the people who responded to the study's recruiting had children with moderate to severe ASD; a few of the respondents had more than one child with autism spectrum disorder. Most of the study participants described themselves as middle class, although the study included some lower-income participants who were members of an autism support group in the New England city.

Recruiting occurred primarily through secular ASD support programs. The majority of the respondents had a Catholic or Protestant background, with a small number of Jewish participants and 13 respondents who reported no religious affiliation. Attempts to recruit Muslim participants were not successful; recruiting of Muslim participants was attempted by recruitment calls and flyers to local Muslim organizations, as well as through personal contacts. An internet survey posted on a Muslim parenting website yielded little response. A table summarizing the demographics of the respondents is presented below (Table 1).

**Table 1.** Demographic and Religious Background of Sample.

| Demographic Background | | | |
|---|---|---|---|
| **Gender** | | **Race/ethnicity** | |
| Female | 48 | White | 41 |
| | | Hispanic/Latinx | 7 |
| Male | 5 | Black | 4 |
| | | Other | 1 |
| **Geographic Region** | | **Marital Status** | |
| New England | 44 | Married | 41 |
| Other | 9 | Separated/Single/Divorced | 12 |
| **Religious Background** | | | |
| **Religious Affiliation** | | **Frequency of Attendance** | |
| Catholic | 26 | weekly | 22 |
| Jewish | 2 | | |
| Episcopalian | 1 | Once a Month | 4 |
| Protestant (unspecified) | 2 | | |
| Methodist | 2 | Infrequently | 13 |
| Pentecostal | 2 | | |
| Presbyterian | 1 | | |
| United Church of Christ | 2 | Never | 14 |
| Lutheran | 2 | | |
| No Affiliation | 13 | | |

The constructs for this study focused on parents' experiences with religion and spirituality vis-à-vis their child(ren)'s ASD. A team of four researchers conducted one-on-one in-depth interviews of people who responded to the study recruitment materials. Before the interview began, the participants were informed of the study, signed consent forms, and filled out a short demographic background survey. The interviews were conducted using a semi-structured protocol and lasted between one and two hours. The interviews were designed to elicit information about parents' experiences with spirituality and religion as they pertained to their child. The interview questions were drawn from relevant academic literature, as well as guided by the experiences of two members of the research team who had young adult siblings with ASD. The questions in the research protocol fell into three major sections. The first group of questions pertained to background information and initial experiences with diagnosis. The second group of questions also pertained to background information about the respondents' experiences with family members, teachers, and medical professionals in parenting a child with an autism spectrum disorder. The third group of questions were the focus of the study, probing the respondents' relationship with religion and spirituality vis-à-vis ASD. This group of question was split into two subsections: (1) religious/spiritual background—for example: Were you raised in a religion; if yes, which one? Do you currently practice a religion; if yes, which one? How frequently do you attend religious services? Did you attend any other services in the past but have stopped now? How important would you say religion is to you (very, moderately, little)? Do you have close friends and acquaintances from your congregation, if you attend? Do you pray privately; how frequently, if yes? Do you read the Bible or other religious writings; How frequently, if yes? Do you have any other spiritual practices; if so, describe? Could you call on your religious leader or others in your congregation if you needed help with something; if so, have you?—and (2) experiences with religion and ASD. These were asked as appropriate, depending on how the respondent had answered the background questions. Examples include:

- Do you think your faith is helpful to you in raising a son/daughter with autism spectrum disorder; how, if so?

- Has your faith/or spirituality felt unsupportive or problematic; how, if so (e.g., Do you ever "blame God" or "feel angry with God")?
- Do you have a religious understanding or explanation for your child's disability; if so, explain?
- [For those who attend services] How have other congregation members reacted to your son/daughter at religious services?
- Has your son/daughter been able to participate in religious education programs; if so, explain?
- Why would you like your son/daughter to have religious faith/instruction (if person does)?
- How can religious communities best meet the needs of children with autism spectrum disorder—i.e., what advice would you give to religious communities?

The interviews were all fully transcribed verbatim, yielding over 1800 pages of narrative data in the transcripts. The interviewers also wrote field notes after the interviews to provide context to the transcripts. When each transcript was complete, the member of the research team who had conducted the interview read the transcript to ensure that it was accurate and complete and also reread the field notes. Two researchers then read each transcript carefully and manually coded the data for key themes, seeking to understand the meaning of parents' experiences within the context (Maxwell 2013). The analytical approach was informed by grounded theory (Corbin and Strauss 1998; Glaser and Strauss 1967), where instead of hypothesis testing, the goal was to find and analyze themes that emerged from the data (Saldaña 2016).

The data analysis was triangulated by two members of the research team, reading and coding transcripts and finding themes emerging inductively from the qualitative data. Saldaña (2016) states that in "themeing the data," themes serve to organize a group of repeating ideas. Themes can consist of ideas or statements from the interview respondents, descriptions of behavior, or explanations of what is happening. A theme "brings meaning and identity to a recurrent experience and its variant manifestations" (Desantis and Ugarriza 2000). Themes allow a deep understanding of the lived experience or lifeworlds of the people interviewed, and "themeing the data" is particularly appropriate for research that aims for this understanding (Saldaña 2016; van Manen 1990). According to van Manen, themes provide "the study of the lifeworld–the world as we immediately experience it pre-reflectively rather than as we conceptualize, categorize, or reflect on it ... Phenomenology aims at gaining a deeper understanding of the nature or meaning of our everyday experiences" (van Manen 1990, p. 9). This data analysis approach fit well with the outcome of the study, which was to gain a deeper understanding through careful qualitative research of experiences with participation in religious education. Saldaña (2016) describes the analysis method of "themeing the data" as using larger passages of data (sentences) instead of short codes to better capture the essence of the participants' meaning.

In analyzing the data, two researchers read and reread through all the transcripts, noting themes that repeatedly emerged from the data. Lines of data with surrounding context, describing the participants' experiences, feelings, and interpretations, were copied and pasted into documents created for each theme as suggested by Saldaña (2016) for this method of qualitative analysis. The analysis was iterative; as the transcripts were read and reread, new themes were added, and later, similar themes were merged. Some pieces of text contained more than one theme, and these were coded as such in all appropriate documents. The themes were cut and pasted using word processing software guided by Saldaña (2016), who argues that, while themeing can lend itself to selected qualitative data analysis programs, "themes are also intriguing to simply "cut and paste" in multiple arrangements on a basic word processor page to explore possible categories and relationships" (p. 180). As themes emerged from a thematic pattern analysis, the research team members read back through the transcripts to ensure that all the data pertinent to the theme had been recorded and appropriately categorized. Conversations between both authors clarified any discrepancies or differences regarding the final list of key themes. Researchers cross-checked each other to ensure accuracy. Theoretical saturation was reached when the coding of transcripts revealed no new or overlooked themes. In addition to "themeing the data," the research team created a large spreadsheet with demographic and religious background information about each participant, drawn from the short demographic questionnaire administered prior to the

interview, as well as from information found in the transcripts. The spreadsheet contained information such as age of focal child, presence and age of other children, marital status, occupation, education, socio-economic status, religious background, frequency of service attendance and private religious practices, self-described salience of religion, and whether they had sought religious education for their children.

In addition to qualitative analysis of in-depth interview data themes, this project included a field visit. Both authors visited an adaptive religious education program designed specifically for children with intellectual and developmental disabilities such as autism spectrum disorder, attending a religious education session and adaptive service and interviewing the trained volunteer religious educator. This visit lasted from three to four hours, including time spent speaking informally with parents at the end of the religious education session. The goal of this brief visit was for the researchers to immerse themselves into an adaptive religious education class designed for children with intellectual and developmental disabilities and to garner feedback from the religious educator and parents present.

## 4. Results

Parents who themselves valued spirituality and religion attempted to share these values and beliefs and transmit them to their children. These parents believed that religion is beneficial to their children and youth with autism spectrum disorder. Parents provided religious socialization in the home by talking to their children about faith and teaching them about prayer, as possible. Many of these parents also wanted to involve their children in congregational participation and religious education, wanting their children to belong to a congregation with their family and obtain religious education. A total of 24 of the 53 participants had tried formal religious education at a congregation for their child with autism spectrum disorder. Of the 29 who did not attempt congregation-based religious education, three of the focal children attended a religious school and thus would not be enrolled in congregational religious education. The 26 remaining participants had a variety of reasons for not attempting religious education for their child with autism spectrum disorder. Some of these families were not themselves religiously active. However, other religiously involved parents believed that their child would not benefit, that it would be too difficult, or that it would be beyond the ability/capability of their child to attend or understand religious education. Several parents specifically stated that they did not believe that it was necessary for their child's faith development: "*She doesn't need to go to classes, which she won't understand, to have a relationship with God*" (female; United Church of Christ; attends regularly; at the time of the interview, her daughter with ASD was 17).

Parents had mixed reactions to the religious education experiences of their children with autism spectrum disorder, with slightly more positive response than negative response emerging from the data analysis. Of the 24 respondents who had pursued congregational religious education, eight had negative experiences, 11 reported positive experiences, and five were non-committal about the experience. A third who recounted their experiences of religious education, however, felt lack of welcome and support, and others did not try to obtain religious education, meaning that the majority of the sample did not report positive experiences. The following sections lay out and illustrate the main themes, analyzing the lived experiences of parents. Pseudonyms are used for all the names.

*4.1. Negative Religious Education Experiences (Eight Respondents)*

Eight parents who had tried to obtain religious education for their children reported negative experiences. Parents described a number of factors contributing to their unsatisfactory experience. Sometimes the experiences were so unsatisfactory that parents left the particular congregation or experienced struggles with their faith. Parents recounted children being socially ostracized, unaccommodating religious educators, age-inappropriate placement, and lack of aides and supports.

### 4.1.1. Social Isolation

Similar to other social situations, participation in religious education classes requires peer interaction. Parents described religious education as an area in which their child was socially ostracized: "*We would find Ben [at religious education class] sitting at a group table coloring. No one would want to sit with Ben; he was socially ostracized*" (female; Catholic; occasionally attends; her son with ASD was 19). One mother recounted her son's social ostracization during textbook lessons on living out religious faith: "*Having middle schoolers sitting at a table purposely avoiding and shunning the outcast and having lectures in the classroom about how to act like Jesus and be Christian. And looking at textbooks. They had this amazing opportunity to teach in real life about how to deal with the downtrodden, and they didn't see it*" (female; Catholic, occasionally attends; son with ASD was 18). In addition to being left out from peer group interaction, some of the respondents' congregations placed their children with ASD in age-inappropriate classes, such as this mother who recounted how her seven-year-old son was kept in a toddler class. "*They were going to keep Collin down and put him in a toddler class ... I said, 'Why did you put him in a toddler class; he is not a toddler' ... They said, 'We just thought it would be best to suit his needs,' and I said, 'That is not best, and he is going to know. He cannot be with the toddlers forever; He is not going to be 15 in a toddler class. He needs to move up with the kids his own age'*" (female; non-denominational Christian; attends regularly; her son with ASD was 7). Thus, social interaction, which is particularly challenging for children and adolescents with autism spectrum disorder (Biggs and Carter 2016), was a focus of some of the problems that parents had experienced in trying to secure religious education for their children. Whether children or youth were ostracized by peers or placed with much younger children, religious education proved an unsatisfactory social experience. This is significant, as Biggs and Carter (2016) found an association between religion and higher youth quality of life for youth with autism spectrum disorders. While these authors posit that religious youth with ASD may have more opportunities to develop friendships with peers in their religious communities, these interviews cited above show that some children remain socially isolated from their peers in religious education.

### 4.1.2. Unhelpful Religious Leaders and Educators

In addition to difficulties with children's peers, several participants related experiences with unsympathetic pastors or religious educators or, at the least, religious educators who did not know how to deal with children with special needs. "*One time I went in and the teacher just had her coloring the whole time. I was like, 'she can participate; she can answer'*" (female; Catholic; attends regularly; her daughter with ASD was 16). This type of situation can become so difficult for families that they decide to leave their congregations, especially if they believe that the pastor is unsympathetic regarding their needs. "*The church we were at, did not accept his autism ... they would not make accommodations for him ... [The pastor] had this rule that there were no electronics allowed in his confirmation class at all. Billy copes [through] technology. I participated as a Sunday school teacher to get my kids involved with the church and all through Sunday school ... there were no exceptions. It became a power struggle between the pastor and my son ... [W]e ended up leaving. It's been really hard; we tried so hard. I'm so heartbroken*" (female; Pentecostal; attends weekly; her two sons with ASD were ages 18 and 23). Thus, the role of institutional leadership is highlighted through these findings.

A number of the respondents expressed their frustration with the low expectations that religious educators had for their children. These volunteer religious education teachers, unknowledgeable about disabilities, demonstrate an implicit and perhaps explicit bias against children with disabilities. Placing children with autism spectrum disorder with toddlers or setting them aside to color instead of participating in class, shows biases leading to low expectations. Parents who encounter religious leaders who are unsympathetic to meeting the special needs of their children become discouraged, sometimes to the point of leaving for another congregation or leaving organized religion altogether. These qualitative descriptions of negative experiences illuminate previous findings from some survey-based studies: that participation in organized religion can increase maternal distress with respect to children

with ASD (Ekas et al. 2009) or that dissatisfaction with support can lead to switching congregations (Ault et al. 2013a).

*4.2. Positive Religious Education Experiences (11 Respondents)*

Despite the negative experiences just described, 11 respondents recounted positive experiences with religious education. These parents mentioned accommodating programs, understanding religious educators, and children who were integrated and supported. It should be noted that some of these parents had changed congregations, educated their congregations about supporting children with autism spectrum disorder, or otherwise actively sought out places where they felt their children would be supported.

4.2.1. Open to Meeting Child's Needs

A handful of parents were pleased with the comprehensive support from their congregation's religious education programs and the level of involvement they were allowed to have in their child's religious education. While previous research on parents with a broader variety of special needs in their children found that parents react negatively to being asked to serve as their child's de facto aide in a religious education setting (Ault et al. 2013b), some parents in this study found it to be positive. "*They let me go in and help ... I am the assistant on Sunday mornings, so I go in with him so it's like I am his one-to-one aide. They have always been really good about 'come on in' and they ask me to give them any tips or any constructive criticism ... they tell me 'we're going to try to do it' I wish the public school was as good as Bible class!*" (female; nondenominational Christian; attends weekly; her son with ASD was 7). The parents seemed to particularly appreciate when their faith communities were willing to work with them to make needed changes to support their children's religious education. "*I have gone to them ... I have said that I know my son is a challenge ... what can we do about that? The church community has come back with options: either I could go with him, or they can put a second teacher in there ... so they do work with us*" (female; Presbyterian; attends weekly; her son with ASD was 10).

4.2.2. Supportive Religious Leaders and Educators

In recounting their experiences with religious education programs, several parents mentioned how they found teachers to be extremely accommodating and to have deep connections with their child. The respondent below has a non-verbal young adult son with severe autism spectrum disorder; she had previously left an unaccommodating congregation before finding a place that would support her son. "*His Sunday school teacher that he's blessed to have ... she loves him very much and understands him very well. She's been with him for at least six years now, she believes that he understands religion in his heart ... we're really blessed to have her*" (female; Coptic Orthodox; attends weekly; her son with ASD was 21).

Some of the respondents also mentioned the openness of religious leaders to including their child in religious education. "*Our Rabbi has been very open and accepting ... I think a lot of it has to do with the fact that he is openly gay, so inclusion is important to him ... we did have to do some education around disability with him*" (male; Jewish; attends occasionally; his son with ASD was 16). The parents also praised volunteer religious education aides to their children. "*When he was younger, a lot of our churches provided a helper for him which was key. I wish we would've had more of a peer helper as he got older ... He's always been integrated in, which I think has been good*" (female; Lutheran; attends weekly; her son with ASD was 18).

These parents' lived experiences point to important factors in embracing children and youth with autism spectrum disorder in religious education as a way to engage them in their congregations and contribute to higher quality of life and positive youth development. Welcoming religious education teachers, congregations willing to provide aides, accepting pastors willing to work with families—these qualities made parents appreciate religious education as a positive for their children.

Slightly over half of the study respondents had not pursued religious education for their child. These respondents had neither positive nor negative experiences to report. Some of this group consisted of people who believed that their child did not need religious education or would not benefit from it.

*"I know people who have sent their kids to religious education ... but I just have never done that. God made this boy autistic . . . he has been baptized . . . surely he will let them into heaven when he dies. I don't feel like he needs to go to catechism . . . If God won't let Austin into heaven by being a disabled person, who will he let in?"* (female; Methodist; rarely attends; her son with ASD was 20). Moreover, in this group were people for whom religious institutional participation was not important themselves, as well as, perhaps, people who felt that religious education would be too much effort given their already stressful lives (Ault et al. 2013b; Whitehead 2018). Several of the respondents suggested that they would like to pursue religious education at a later time or if they found the right congregation.

These findings suggest that despite the respondents who had positive experiences to share, work remains to be done for congregations to make religious education available to this population. If the increased social capital of youth involved in religious organizations contributes to positive youth development (in the literature on typically developing youth) and higher quality of life (in the literature on youth and young adults with autism spectrum disorder), then it is imperative for congregations to provide religious education programming that is supportive and welcoming.

### 4.3. Field Visit to an Adaptive Religious Education Program

While a few of the respondents recounted religious education programming in which teachers had made individual accommodations, very few parents mentioned in the in-depth interviews any sort of official religious education programming specifically designed for children with disabilities. Roebben (2012) argues for inclusive religious education where "people with and without special needs learn in each other's presence ... radical inclusive religious education ends up with the experience of learning as 'receiving the gift of friendship'" (p. 1175). Some parents prefer and seek out specialized programs.

Religious education programs focused on those with intellectual and developmental disabilities are not abundant, but they do exist. One example is a Roman Catholic program that operates in a number of areas in the United States, called Special Religious Education programs, known as "SPRED." Both authors of this article made a field visit to a SPRED program, in addition to interviewing the religious educator and informally conversing with parents present. According to the program's website, SPRED exists to help congregations integrate people with intellectual and developmental disabilities "into parish worship through the process of faith development" (Roman Catholic Diocese of Worcester 2018). This is accomplished by providing training courses for religious educators, providing materials for the religious development of children with intellectual and developmental disabilities, and providing observation opportunities for parish leaders, families, and prospective volunteers to learn about faith development and congregational integration for these children. They also provide regular family liturgies for families of the participants (Roman Catholic Diocese of Worcester 2018).

The SPRED site visited as part of this research project has been operating for almost 10 years in a suburban Catholic parish. There were five children in the session on the day of the site visit, all of whom had ASD that appeared to substantially impact functioning. The director at the SPRED program in this parish noted her belief that the family is the first community a child knows, and therefore, if children are to have a relationship with faith, it is important to make a connection between the family and the congregation. Parents are welcome but not required to participate in SPRED sessions.

According to the volunteer SPRED religious educator, every aspect of the environment is chosen and organized with great care. The rooms used are meant to indicate a soothing home-like setting, trying to reach all senses to help the children participating feel calm and centered on faith. Photos adorn the walls of the space—visual reminders of how to make the sign of the cross and photos of important Biblical figures. The director pointed out the special dim lighting, which gives each room a peaceful glow. Fresh flowers everywhere add to the overall ambience. Children in the class are referred to strictly as "friend". SPRED sessions last for approximately two hours and children typically attend classes every other week. Sessions consist of activities related to prayer and developing an understanding of faith concepts through a variety of methods. At the end of each session, everyone meets in the celebration room, a quiet room with candles and soft textiles on the wall, to participate in a

small religious service before a snack is served. One goal for the majority of the children in the program is to make their First Communion at some point, a goal usually accomplished with the combined efforts of catechists and parents alike.

In informal conversations after the end of the session, parents told the authors how much this program meant to them. They stated that they were very pleased at the ways in which their children with ASD were supported, taught, and reached; some had come to this parish specifically because of this program. It should be noted that this group of parents represents a subset of parents who prefer specialized religious education for their children with autism spectrum disorder. Other scholars, practitioners, and families have advocated instead for an inclusive model (Roebben 2012), where children and youth have an opportunity to make friendships with non-disabled peers.

For the particular parents at the SPRED session visited for this study, the adaptive religious education was praiseworthy. These types of programs, for some parents, demonstrate care, sensitivity, and informed knowledge regarding what reaches children with autism spectrum disorder. The tactile emphasis, soothing environment, and varied activities are designed to center the needs of children with intellectual and developmental disabilities. Volunteer teachers, similarly, have received specific training regarding how to impart religious education to children with intellectual and developmental disabilities. In addition to being perceived as beneficial to their children by parents who choose this, this type of adaptive religious education is viewed as demonstrating that religious communities notice and care about children with these disabilities.

### 4.4. Recommendations for Congregations

One of the final questions on the semi-structured interview protocol asked: Do you have any advice or recommendations for religious organizations? The list of recommendations below was generated from parents' responses and other findings in the data.

1.　Far and away the most frequent recommendation was to have people in religious congregations trained to work with children with developmental disabilities. As one parent said, speaking of religious educators:

   *They should all go through a course. just a standard course of acceptance . . . I think some of them were nervous . . . They did not know what she was capable of or whatever or what she could or could not do, so they just backed away and did not put in the effort. It's frustrating, because you kind of constantly have to step in an advocate.*

   Others echoed the advice:

   *I would like churches and synagogues to just have specialists on hand; you can include these kids just the way public school does.*

   *Having somebody trained to take care of special needs kids.*

Training teachers would help avoid the negative religious education experiences described by the respondents in the study, such as age-inappropriate placements with much younger children or low teacher expectations of children with disabilities. Yet, even as parents recommended more training for people working in religious education, some recognized the difficulties inherent in doing so:

*The level of expertise a person would have to have is that would be a tremendous volunteer or paid person. You have to have some ABA training. You have to have autism training. You have to know how to deal with difficult behaviors. If today, if he were to go somewhere today, they would have to know all the tricks of the trade to get compliance from him and work through it, and it would have to be the same person every time, because you need that consistency. That's a lot to ask at one person for an hour, once a week.*

*Well, like I said, we do offer that autism course once a year, and I wish that it would be required to go, but it's [sic] volunteer position so you cannot require them to do anything. So, I wish they were better educated, but they do have their own families and their own commitments.*

Most religious educators are volunteers with busy lives, often the parents of their own children in the religious education classes. On top of whatever work, parenting, and other commitments in their lives, they are trying to teach religious education material to a roomful of children. Without training, it is no wonder that they are ill-prepared to reach students with autism spectrum disorder and other developmental disabilities.

2.  Make aides available for children to better enable them to participate in religious education. While training of teachers was a high priority, placing aides in classrooms is another recommendation, according to parents/caregivers. One young adult sibling described the buddy system which had enabled her sibling to participate in religious education; she herself had served as another child's buddy as an older teenager.

    *They sort of have their own Best Buddies program, (Belongings of Religious Experience?). It's called the BRE council ... People [with] special needs, they all match with the buddy, to sit with them, if they need to leave or go out with them.*

Other parents also described how aides had helped their children successfully engage in religious education.

*When he was younger, a lot of our churches provided a helper for him, which was key.*

Some parents welcome the opportunity to work with their children in religious education ("*They have always been really good about 'come on in', and they ask me to give them any tips or any constructive criticism*"). Others resent the assumption that they should be their child's personal aide in order for the child to participate in religious education (Ault et al. 2013b). Religious organizations should make aides available and afford parents the preference of either using the aide or participating with the child themselves.

3.  Be aware that different parents have different preferences regarding inclusion for their child. The first two recommendations assume a model of inclusion, and indeed, many parents do want inclusion in regular religious education.

    *I think the world should accept him as he is without making him go to a special Sunday school with other people that have disabilities. Why can't they come to us? Why do we have to have a special program? I hate that. I want him to be included. And maybe that's stupid, maybe that's a pipe dream. My dream is to have churches let them be with us and embrace that. I don't send him to the special needs prom. He went to prom. I find it exclusionary. I think churches should, from beginning to end embrace differences and make whatever accommodations need to be made to allow people with differences to be there.*

However, some parents prefer a model of specialized instruction with other children with disabilities, as in the parents at the field visit program described earlier. In addition to religious education, some parents in this study wished for a religious service that would be comfortable for people with disabilities. "*Maybe have a specific mass for people during the week that is a comfortable place for people with disabilities ... A place where you can go, and you can be okay, and you don't have to worry about everyone staring at him.*"

Overall, a best-case scenario is when a religious congregation itself is educated and accepting of people with disabilities. One father said that while he and his spouse "*had to do some education around disability*" with the rabbi and members of their temple, it is an inclusive place where his son (who participated in religious education and made his bar mitzvah) is accepted:

*I think the primary benefit for him is that it is a place where he can be expected to be accepted ... At the risk of getting more choked up about it ... I think early on, his behavior was such that it was hard for him to participate in a lot of community events, so we joke about how he was probably thrown out of every child activity in town. There was a story hour at the local bookstore, and he couldn't go to do that—the woman asked us not to bring him back. He could not participate at this group at the YMCA. There is a lot of stuff in town where he couldn't be a part of. The fact is that he can be at the temple, and he can participate.*

While the respondent quoted above had the energy and cultural capital to educate his religious congregation on disability to positive effect, this burden should not be placed on already over-burdened parents.

4.  Notice, welcome, and support families of children with disabilities.  This is a blanket recommendation that draws together a number of others. This could include support groups, as several parents recommended. From one mother who had stopped attending church, because it was too stressful to bring her son: *"They should know who among in the families in the congregation have a child with autism and have some sort of a support network among those families."* Behind the specific recommendations for programs or supports, parents plead for welcome. *"If you say that all are welcome in this place, being church, you make it that all are welcome in this place and you embrace diversity on all levels . . . If you say that all are welcome, you need to welcome all." "We have to accept these children into our community. I would give advice to accept them." "[Religious congregations] have got to make more effort to reach these families."* In short, as one parent said, *"Church should not be another place of struggle."*

## 5. Discussion

The results of this study add to the very few existing qualitative studies of religious engagement and socialization of children and youth with autism spectrum disorder specifically focusing on religious education. This study adds insight gained from in-depth, face-to-face interviews and illustrates factors affecting both positive and negative experiences with religious education. As research suggests that religious participation can positively impact quality of life and development for children and youth with ASD (Biggs and Carter 2016; Liu et al. 2014), this study's findings are significant.

There are several key contributions of this study.  There is currently a paucity of qualitative research focused on religious education experiences of children with autism spectrum disorder from the point of view of parents.  This study aims to fill this gap.  Its in-depth interview methodology, as opposed to survey research, illuminates the experiences of parents/caregivers. This methodology allows parents' voices to be clearly heard and enables a deeper understanding of their experiences with religious education. In another contribution, this study finds more evidence of positive religious education experiences than some previous studies, as, of the 19 respondents for whom it could clearly be determined, slightly more had positive experiences than negative ones. With supportive religious leaders and trained and flexible religious educators who are comfortable and skilled in working with children with autism spectrum disorder, religious education was a positive part of children's lives.  Similarly, unlike previous research, parents in this study did not bring up resenting being expected to be their child's personal aide in order for them to participate in religious education; parents for whom this was the case had positive experiences to report.  Some parents mentioned that they had needed themselves to do work in educating their religious leader and/or congregation about supporting children with autism spectrum disorder—an extra form of emotional labor that could potentially be negative but, in this data, was not mentioned as so. It should be noted that some of the parents who reported positive experiences had switched congregations before finding one, actively sought out congregations that would support their children, or provided education on disability to their congregation. This is a level of extra work for parents already facing challenging circumstances. Furthermore, numerous parents had negative experiences and other parents felt deterred from trying

religious education for their children, meaning that most parents in the study did not report positive experiences. Parents had encountered inflexible or unaccommodating religious leaders, religious educators unknowledgeable about working with children with disabilities, classroom settings where children with autism spectrum disorder are ostracized, and age-inappropriate placements. For some, their frustrating experiences caused them to leave a particular faith community in search of one more open to meeting their children's needs or to stop participation in organized religion altogether.

Thus, another contribution of this study is recommendations for congregations, which are drawn directly from parents' voices. Finally, this study expands the literature on positive youth development by using a lifespan model to include children and youth with autism spectrum disorder. These youth are rarely considered in research on religion and positive youth development.

Slightly over half of the parents interviewed had not sought religious education for their child(ren) with ASD, either believing that the children did not need it or noting that they themselves were not actively involved in organized religion (sometimes because of their children's condition). Although unstated in this data with regard to religious education specifically, parents may simply also be tired of battling for their children in numerous arenas and see religious education as one more difficult thing to be managed. Speaking about congregational participation more generally, one respondent noted, *"Church should not be another place of struggle. Like school can be a struggle, or going to the store can be a struggle, or getting along with your family can be a struggle, but church should not be not be a place where you struggle. I think the vast majority of people whose kids are on spectrum are not in church because it is another place of struggle."*

The American Association on Intellectual and Developmental Disabilities' policy paper on spirituality advocates for accommodations that would enable people with disabilities to participate in spiritual or religious activities and encourages faith communities to develop strategies of welcome and inclusion (AAIDD/The Arc 2010). Characteristics of inclusive faith communities include faith leaders committed to inclusion, positive portrayal of people with disabilities, ties with disability organizations, and use of educational resources on disability (Griffin et al. 2012). Working against inclusion may be negative or condescending attitudes of religious leaders and members, narrow ways in which information is communicated, and programmatic barriers (Carter 2007).

Numerous parents of children with autism spectrum disorder want their children to have religious education, believing it to be beneficial to their children and important in passing on family values and traditions. Some of the respondents in the current study brought up their hope that religious education would contribute to giving their child more social interaction and helping their child be part of a community. The fact that a significant number in this study either had negative experiences or perhaps felt discouraged from trying points to more work needing to be done by religious organizations. In addition to considering religious socialization and its impact on positive youth development, it is important to remember that interactions in congregations impact parents' mental health. Research studies attest to the positive mental health benefits (higher life satisfaction, sense of well-being, increased sense of social support, etc.) associated with participation in supportive congregations and the increased distress associated with negative interactions in congregations (Ekas et al. 2009; Ellison et al. 2009; Lim and Putnam 2010; Patrick and Kinney 2003).

The existing limited research studying the impact of religion on youth and young adults with intellectual and developmental disabilities including autism spectrum disorder finds an association with positive youth development. Religiously involved young adults or parents of youth with ASD report that religion promotes more peer social support and increased social capital, opportunities for youth volunteering, and a personal resource for help and self-acceptance (Biggs and Carter 2016; Liu et al. 2014). Religion is not important for all people, but it can be an important resource for some that leads to positive outcomes. If children and youth are not provided with appropriate and welcoming religious education, then they lose an important way to participate in their religious organizations.

## 6. Conclusions

A neurodiversity perspective views autism spectrum disorder as a form of diversity akin to other types of diversity (Nicolaidis 2012; Robertson 2010). This highly prevalent type of diversity has been neglected in research on religion and positive youth development. Clearly, continued academic research is necessary on (1) the impact of religion on positive youth development and/or quality of life for youth with ASD and (2) ways in which children with ASD can access and participate in religious organizations and programming, including religious education, for families and youth who desire this. This article finds that some families have positive experiences for their children through open and supportive religious leaders, welcoming congregations, and accommodating educators. Others find religious educators unknowledgeable, unwilling, or unable to work with special needs children, social isolation for their children, or unhelpful religious leaders.

### 6.1. Recommendations for Congregations

One contribution of this study is recommendations for congregations from parents/caregivers regarding religious education. These recommendations are detailed earlier in the article and summarized again here. First and foremost, parents want congregations to have people trained to work with children with developmental disabilities such as ASD. Training could help avoid situations such as age-inappropriate places or low teacher expectations. At the same time, parents recognize that religious educators are volunteers with busy lives of their own, such that adding training for working with children with ASD might be difficult. However, Baggerman et al. (2015) report that volunteer religious educators could successfully be coached on how to work effectively with children with moderate to severe disabilities in an inclusive religious education class. Parents also request that congregations make aides available to better help their children to participate in religious education classes. While some parents voice positive experiences of serving as their child's own aide, some parents resent this being expected of them (Ault et al. 2013b), and congregations should make other aides available. Congregations should also be aware that some parents prefer religious education where their child with ASD is included with non-disabled peers, and other parents prefer specialized programs. All congregations should have volunteer teachers trained and aides available for integrated programs, where children with ASD can interact and develop relationships with typically developing children. Congregations that do offer specialized programs should publicize their offerings to local ASD support centers so that parents who seek this type of program can find these congregations. Finally, parents of children with ASD plead for congregations to notice, welcome, and support them.

### 6.2. Recommendations for Future Research

The limitations of this study include the demographic background of its participants, 77% of whom are white and self-described as middle class, although lower income participants and people from diverse racial and ethnic backgrounds did participate in the study. The study moves a bit from the Protestant Christian focus of much existing research in that the majority of the sample who reported a religious affiliation were Catholic, with two Jewish respondents. However, efforts to recruit Muslim respondents were not successful. Future research must include Muslim participants and other underrepresented religious groups. In general, further research must include more racial, ethnic, religious, and geographic diversity. Nationally representative surveys that include more detailed information about religion and spirituality in the lives of children and youth with ASD are needed. Similarly, more extensive qualitative research that covers as large a number of respondents as possible, from diverse religious, racial, ethnic, socioeconomic, and geographic backgrounds needs to be conducted. Despite limitations, this study makes an important contribution, taking a lifespan approach to study the religious socialization opportunity of religious education for children and youth with autism spectrum disorder.

This study points to a pressing need to expand research on the role of religion in positive youth development to the population of youth with autism spectrum disorder—a noteworthy number of youth in the United States. Although one line of research suggests that some people with ASD are less likely to hold religious beliefs (Norenzayan et al. 2012), research discussed earlier in this article finds religious involvement to be beneficial for some youth with autism spectrum disorder. Whether conceptualized in the traditional constructs of positive youth development (perhaps for youth with higher functioning ASD) or quality of life (perhaps for youth more profoundly disabled), this is a population that has been all but overlooked in this area of research. Research along the line of Biggs and Carter (2016) quantitative study on the role of religion in quality of life for youth with ASD and Liu et al. (2014) small qualitative study of the impact of religion on the lives of young adults with intellectual and developmental disabilities would further flesh out the mechanisms through which religious participation can contribute to the lives of youth with autism spectrum disorder. At the same time, recognizing that religion is not always a positive factor in youth development (Abo-Zena and Barry 2013; Magyar-Russel et al. 2014; Page et al. 2013), it would be important for future research to consider if and how religion contributes negatively to youth development for youth with autism spectrum disorder. If youth experience social isolation in religious education or age-inappropriate placement, social isolation or rejection in congregations, or other negative experiences, how does this impact youth development?

A lifespan approach on religion and youth development suggests that another area for more academic research is religious education of children and youth with autism spectrum disorder who belong to diverse religions and ethnicities. There is a growing amount of general literature on religious inclusiveness of people with ASD, including books such as *Autism and Your Church: Nurturing the Spiritual Growth of People with Autism Spectrum Disorder* (Newman 2011). One resource is "Autism and Faith: A Journey into Community" (Walsh et al. 2008), a booklet that writes about inclusive religious practices for people with ASD. Although not about religious education in an institutional environment such as a mosque-based setting, Jegatheesan et al. (2010) ethnography of three South Asian immigrant families shows Muslim families teaching children with ASD daily ritual prayers in Arabic and bringing them to participate in the mosque. There are resources specific to particular religious traditions, such as the Adaptive First Eucharist Preparation Kit: For Individuals with Autism and Other Special Needs (Rizzo and Rizzo 2011) or Nes Gadol, a multisensory, individualized Bar/Bat Mitzvah training program designed for children of all abilities, including non-verbal children (Hall 2015). The National Catholic Partnership on Disability convened an Autism Task Force from 2010 to 2014 to assess programming and curriculum for individuals with ASD; its work continues under a Council on Intellectual and Developmental Disability (National Catholic Partnership on Disability 2015). Scholars have also provided recommendations from their findings regarding inclusion of people with intellectual and developmental disabilities in religious organizations (Carter 2013). Clearly this is an issue of interest for a significant number of people, given that there are high levels of both autism spectrum disorder and religious practice in the United States. Research needs to continue, with special emphasis on underrepresented religions.

The current study contributes new insight into how children and youth with autism spectrum disorder experience religion in religious education—religion that may contribute to positive youth development. This study fills a gap by contributing to a paucity of in-depth qualitative studies on religious education for youth with autism spectrum disorder. It provides deeper understanding and insight into the lived experience of parents' experiences of trying to provide religious education for their children with autism spectrum disorder—something these parents believe is important and beneficial to their children. The recommendations for congregations provided are drawn directly from parents, as well as an analysis of factors influencing positive and negative religious education experiences. Significantly, in this article, we also highlight the need to extend religion and positive youth development research to the large number of children and youth with autism spectrum disorder, making sure to include diverse religious backgrounds. As the very few existing studies do find a

role for religion in positive youth development in this often-marginalized population, research needs to be conducted and religious organizations need to adopt best practices. One in 59 US children is currently diagnosed with autism spectrum disorder. If religion indeed can contribute to positive youth development in this large number of children and youth, much work remains to be done.

**Author Contributions:** Conceptualization: S.C.S.; methodology: S.C.S.; data collection: V.A. and S.C.S.; writing—original draft preparation: S.C.S.; writing—review and editing: V.A. and S.C.S.; project administration: V.A.; supervision: S.C.S.; funding acquisition: S.C.S.

**Funding:** This research was funded by Covenant Health/St Joseph's College; Faith, Spirituality, and Health Research Grants, Grant: "Faith, Spirituality, Religion, and Autism".

**Acknowledgments:** The authors gratefully acknowledge the research assistance of Mary Bassaly, Samantha Schuetz, and Andrew Whitehead.

**Conflicts of Interest:** The authors declare no conflict of interest. The funders had no role in the design of the study; in the collection, analyses, or interpretation of data; in the writing of the manuscript, or in the decision to publish the results.

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
