# Peer review of "Religion and Positive Youth Development: Challenges for Children and Youth with Autism Spectrum Disorder"

_religions, doi:10.3390/rel10100540_

Round 1
Reviewer 1 Report
This study uses qualitative methods to examine the experiences of children with autism in religious settings by interviewing their parents. It is a strong paper that competently executes and achieves its goals. I offer the following thoughts in the spirit of possible improvement.
Overall, I think the paper is a bit bloated. There are paragraphs where the authors recount another paper's findings over 4-5 sentences. I think this paper would achieve a greater impact if the literature review section was significantly streamlined. The headings seem to be logical. I would urge the authors to try and reduce some of those sections by half. I don't see a need for highlighting the methods, findings, and conclusions of prior papers. Use only what you need to move your own study along.
The authors might consider including one table that breaks out the demographics and religious affiliations of the interviewees. While some of this information is in the text, a table would catch the eye and allow for a much quicker and easier analysis of the sample.
I'm not sure that the ethnographic field visit is necessary or really adds much to this paper. The authors should consider jettisoning that section. Focus on the interviews and those findings.
It seems that the main findings are that people have mixed experiences going to religious communities with their children who are on the autism spectrum. I think the authors might do more to bring out exactly how this adds to what we already know. Or, if there were specific aspects of their findings that were not included in prior research. As is, the manuscript is kind of a summary of all they found, with no real hook.
Finally, the authors should consider, or the editor should recommend, the abstract be shortened considerably. There were whole parts of it that were reproduced from the introduction. It is so long that readers will check out well before they finish it to see what the study is even about. I would suggest it be about a third as long as it is.
Author Response
Thank you for the prompt and helpful review. Below are responses to Reviewer suggestions:
Streamline literature review: The literature review has been streamlined and significantly shortened.
Consider including the demographic information in a table: The demographic information is now included as a table (p.9 on tracked changes manuscript).
Consider jettisoning the field visit: For now I have chosen to keep it in, as it does add a portrait of a particular model of religious education and other reviewers did not suggest removing it. I am happy to remove it if the academic editor of the journal wishes it done.
Highlight main findings: Now the main contributions of the article are more clearly laid out and highlighted at the beginning of the paper (p.2) and summarized in the discussion (p. 16). Drawing on the unique data drawn from in-depth interviewing, recommendations for congregations are clearly provided on pp. 18-20 of the tracked changes manuscript. Recommendations are provided directly from parents interviewed, in their words.
Thank you very much for the careful review comments which have helped to improve the manuscript.
Reviewer 2 Report
“Religion and Positive Youth Development: Challenges for Children and Youth with Autism Spectrum Disorders”
Religions #547026
Much sociological literature has pointed to the significant contribution of religion to positive youth development. However, as this important article argues, this literature often leaves out the experiences of certain youth groups, such as racial minorities, those from lower socioeconomic statuses, and other identities. This article therefore attempts to fill a gaping hole in this research by focusing on children with one of these marginalized identities: autism. The authors note that 1 out of every 59 children in the United States now has an autism spectrum disorder, and that religious instruction can play an important role in supporting their growth. The authors impressively conducted in-depth, face-to-face interviews with 53 parents of children with autism, as well with educators in this field and ethnographic observation of one specialized program. Overall, the authors find that that parents report both positive and negative aspects of their children’s religious experiences. Some believed that religious leaders were accommodating and understanding, which provided their children with support, while others had the opposite experience, noting the lack of appropriate programming and social isolation. Finally, the authors make suggestions for religious educational reform.
Theoretically, this article is extremely strong and makes an important contribution to the literature on the role of religion and positive youth development. The theoretical framework is extremely comprehensive, and notes the variety of ways religious institutions function together to assist youth as they mature and age. Within this context, the authors note that there is relatively little known about religious socialization and children with autism disorders. This population has numerous challenges and this article deftly analyzes the ways that these challenges are either met or left unmet.
Methodologically, the authors impressively have conducted 53 in-depth, face-to-face interviews and used grounded theory methods to analyze their data. My minor suggestions for the authors would be to provide some more detail regarding their methods. As a reader, I wanted to know when the recruitment period started and finished. I would like to know more about the recruitment information that was provided to potential respondents and when the interviews were conducted. If possible, a table with descriptive statistics on the sample would help the reader understand who was interviewed. The interview questions were provided in appropriate detail. Was a particular software program used to analyze the data, and how did the others know that their data were trustworthy? What was the breakdown of those who had pursued religious education versus those who had not? The ethnographic results were interesting and well-described.
Overall, this is an excellent article that contributes to the knowledge that we must have on religious socialization and those children with autism. It concludes by providing highly detailed recommendations for additional research in this area, which will hopefully translate into more meaningful lives for those with this disability.
Author Response
Thank you for the prompt and helpful reivew. Below are responses to Reviewer suggestions:
The draft manuscript now contained requested information about recruitment and interviewing, including details from the material sent to organizations, when the recruitment started and finished, and when the interviews were conducted.
A table is now provided summarizing demographic information (p. 9 of tracked changes manuscript).
Discussion of software consideration for analysis and how accuracy of findings was ensured is now included in the expanded methods section (pp. 7-11 in tracked changes manuscript)
The breakdown of who pursued religious education and who did not is now included (p. 11).
Thank you very much for the extremely helpful review suggestions which helped improve the manuscript.
Round 2
Reviewer 1 Report
This is a fine study. Well done!
Author Response
Thank you very much.